# Prevalence, Distribution and Risk Factors for Trematode Infections in Domesticated Ruminants in the Lake Victoria and Southern Highland Ecological Zones of Tanzania: A Cross-Sectional Study

**DOI:** 10.3390/vetsci11120595

**Published:** 2024-11-25

**Authors:** Godlisten Shedrack Materu, Jahashi Nzalawahe, Mita Eva Sengupta, Anna-Sofie Stensgaard, Abdul Katakweba, Birgitte J. Vennervald, Safari Kinung’hi

**Affiliations:** 1Department of Veterinary Microbiology, Parasitology and Biotechnology, College of Veterinary Medicine and Biomedical Sciences, Sokoine University of Agriculture, Morogoro P.O. Box 3019, Tanzania; nzalawahej@sua.ac.tz; 2National Institute for Medical Research (NIMR), Mwanza Research Centre, Mwanza P.O. Box 1462, Tanzania; 3National Institute for Medical Research (NIMR), Tukuyu Research Centre, Mbeya P.O. Box 538, Tanzania; 4Department of Veterinary and Animal Sciences, University of Copenhagen, DYRLAEGEVEJ100.2, DK-1870 Frederiksberg C, Denmark; msen@sund.ku.dk (M.E.S.); asstensgaard@sund.ku.dk (A.-S.S.); bjv@sund.ku.dk (B.J.V.); 5Institute of Pest Management, Sokoine University of Agriculture, Chuo Kikuu, Morogoro P.O. Box 3110, Tanzania; katakweba@sua.ac.tz

**Keywords:** domesticated ruminants, trematode infection, ecological zone, Tanzania

## Abstract

**Simple Summary:**

Trematode infections (fascioliasis and schistosomiasis) are emerging snail-borne zoonotic diseases worldwide, with a great capacity for expansion due to animal and human movements, climate change, and anthropogenic modifications of freshwater environments. There is a lack of information regarding the prevalence, distribution, and risk factors for trematode infections in domesticated ruminants in Tanzania. This study aims to fill the knowledge gaps by determining the prevalence, distribution, and risk factors for trematode infections in domesticated ruminants in two ecological zones of Tanzania. Fecal samples were collected from domesticated ruminants and examined for trematode infections using the sedimentation technique. The highest trematode infections of *Fasciola gigantica* (*F. gigantica*), Paramphistomidae, and *Schistosoma bovis* (*S. bovis*) were observed in the Lake Victoria zone (Simiyu region) of Tanzania compared to the Southern Highland zone (Iringa region) of Tanzania. To the best of our knowledge, this is the first study reporting trematode infection in domesticated ruminants in the Lake Victoria and Southern Highlands ecological zones of Tanzania. Effective and community-directed prevention and control strategies are highly needed to address trematode infections of domesticated ruminants in these areas.

**Abstract:**

Trematode infections cause long-term suffering and debilitation, posing a significant threat to global animal health and production and leading to considerable economic losses. Studies on the epidemiology and control of these infections in Tanzania are limited. The few available studies have been conducted in abattoir settings. This study aimed to fill this knowledge gap by determining the prevalence, distribution, and risk factors for trematode infections in domesticated ruminants in two different ecological zones of Tanzania. A cross-sectional study was conducted in the Lake Victoria and the Southern highlands ecological zones of Tanzania. Rectal fecal samples were collected and examined for *F. gigantica*, Paramphistomidae, and *S. bovis* infections using the sedimentation technique. A total of 1367 domesticated ruminants were sampled and examined for trematode infections. The overall prevalence of trematode infections was found to be 65.7%. The individual prevalence of *F. gigantica*, Paramphistomidae, and *S. bovis* (based on egg morphology only) was 35.1%, 60.2%, and 3.1%, respectively. Adult cattle were more likely to be infected with Paramphistomidae (adjusted odds ratio, (AOR): 1.98; 95% confidence interval, (CI): 1.40–2.78) and *S. bovis* (AOR: 8.5; 95% CI: 1.12–64.19) than weaners. It was observed that trematode infections in domesticated ruminants are prevalent in the two ecological zones of Tanzania; therefore, effective and community-directed prevention and control strategies are highly needed to address trematode infections of domesticated ruminants in these areas.

## 1. Introduction

Trematode infections, particularly fascioliasis and schistosomiasis, are among the most common helminth infections in domestic ruminants worldwide [1,2]. Adult trematodes are sometimes referred to as “flukes” and the families that contain parasites of veterinary importance include Schistosomatidae, Fasciolidae, Paramphistomatidae, and Dicrocoeliidae [3,4]. The genera *Fasciola* (liver fluke), Paramphistomidae (rumen/stomach fluke), and *Schistosoma* (blood fluke) [5] have similar life cycles, in which domestic ruminants and humans (except rumen flukes) are definitive hosts, while wild animals serve as reservoirs [6].

The presence of these parasites is widespread in tropical and subtropical regions globally, thriving in areas with favorable climatic, ecological, and hygienic conditions [7,8]. In most African countries south of the Sahara, these infections are considered to be endemic [9]. The presence and distribution of freshwater snail intermediate hosts often influence the occurrence of trematode infections [10,11,12].

Fascioliasis is an economically important disease of domestic ruminants, particularly cattle and sheep, and occasionally humans. Bovine fascioliasis is regarded as the most important zoonotic helminthic infection in tropical countries, with the prevalence ranging from 30% to 90% [13]. The two most important species of *Fasciola* parasite that are commonly considered the causative agents of fasciolosis are *F. hepatica* and *F. gigantica* [3]. The two species *F. gigantica* [14] and *F. hepatica* [15] have been widely reported in Tanzania.

There are several genera of paramphistomidae: *Paramphistomum*, *Cotylophoron*, *Calicophoron*, *Bothriophoron*, *Orthocoelium*, and *Gigantocotyle*, of which *Paramphistomum* is the most common and widespread in ruminants [16]. There are several reports of coproscopic and abattoir surveys indicating the widespread nature of Paramphistomidae species and *Dicrocoelium hospes* in Tanzania [17,18,19]. Paramphistomidae are traditionally regarded as having no clinical significance [20]; however, heavy infection with immature flukes, which attach to the lining of the upper part of the small intestine, may cause severe disease, which may even result in death [21].

Schistosomiasis is thought to infect more than 165 million animals in Africa [22]. *Schistosoma bovis*, *S. curassoni*, and *S. mattheei* infect livestock such as cattle, sheep, and goats all across Sub-Saharan Africa [23]. In Tanzania, several reports of coproscopic and abattoir surveys indicate the widespread distribution of *S. bovis* [11,17,24].

In the Southern Highlands of Tanzania, about 100% of the total condemnations of bovine liver on slaughter slabs, poor growth rate, reduced milk production, and infertility are attributed to trematode infections [18,25]. In Tanzania, bovine trematode infections have been reported in all geographical zones by abattoir surveys [26,27,28,29,30]. However, most studies on the epidemiology and control of these parasites in cattle have been carried out in the Southern Highlands [7,14,15,31]. In these studies, the prevalence of fascioliasis and schistosomiasis ranged from 18% to 94% in domesticated ruminants, depending on the production system used [18,25].

Despite these findings, there is a paucity of information regarding the magnitude, distribution, and risk factors for trematode infections in domesticated ruminants in the Lake zone and other areas of Tanzania. This study was therefore conducted to address this gap and generate baseline data that will provide the scientific evidence needed to design effective and locally accepted control interventions against trematode infections.

## 2. Materials and Methods

### 2.1. Study Areas

This study was conducted from March to May 2023 in the Misungwi, Bariadi, and Iringa rural district councils of the Mwanza, Simiyu, and Iringa regions, respectively. Within Misungwi district, the study was carried out in Kanyelele, Koromije, and Ibongoya B villages. In Bariadi district, the study was conducted in Pugu and Itubukilo A villages. On the other hand, in Iringa rural district, the study was conducted in Lupembelwasenga, Usengelindeti, and Migori villages (see Figure 1). The study districts and villages were selected purposively based on the available domesticated ruminant population, disease history, and disease ecology described in routine reports from the respective local District Veterinary Offices.

The Bariadi and Misungwi districts are located in the northwestern part of Tanzania to the southeast of Lake Victoria. Both districts experience a low rain season from October to December and a high rain season between March and May of each year, while the dry period usually runs from January to the end of February and from June to the end of September of each year. The districts receive 700 mm to 950 mm of rain annually [32]. The temperatures in the two districts range from 190 °C to 290 °C [32].

The Iringa Rural District Council is located in the Southern Highlands of Tanzania. The highland area in the eastern part of the district consists of many hills and valleys, along with numerous permanent rivers, streams, and ponds. In contrast, the flat lowland area on the western side is semi-arid and characterized by dry grazing land with thickets and scattered bushes, as described by Mahoo [33]. The annual rainfall in highland areas ranges from 500 mm to 2700 mm, while in lowland areas, the rainfall is less than 600 mm [31].

### 2.2. Study Design and Sampling Procedure

This study was a cross-sectional survey whereby rectal fecal samples were collected by trained veterinary assistants from cattle, sheep, and goats randomly selected from household farms in the study villages. Household farm-level information such as the household name, sub-village, and flock size was recorded. Individual-level animal information such as the age (classified as weaners and adults), sex, breed (local-breed and cross-breed), and body condition (lean, median, fat) was also recorded. Unique identification numbers and dates were used to label the samples, after which they were stored in an ice-packed cool box and transported to the laboratory of the National Institute for Medical Research (NIMR), Mwanza Centre, for laboratory examination.

The sample size calculation was performed using the following formula: N = (Z^2^ * *p* * (1 − *p*))/d2, based on an expected prevalence (*p*) of bovine fascioliasis of 33%, as reported by Nzalawahe et al. [34], and a desired absolute precision (d) of 3%, with a 95% confidence interval (CI) as per Thrusfield’s study [35]. A total of 1367 domesticated ruminants (cattle 739, goats 319 and sheep 309) were sampled in all the study districts and examined.

### 2.3. Coprological Examination

Fecal samples from domesticated ruminants were processed using the fecal sedimentation method [36]. In brief, approximately 10 g of fecal material was placed in a plastic container and mixed with 30 mL of tap water. Using 500 mL conical beakers, the fecal suspension was sieved through a wire mesh with an aperture of 250 μm and then filled with tap water. The suspension in a beaker was then allowed to settle for approximately 15 min.

Thereafter, the process of decanting the supernatants and re-suspending the sediments in tap water was repeated three times. Eventually, all the sediment was placed into a Petri dish, stained with 1% methylene blue, and observed under a light microscope at 10x magnification for examination of the parasite eggs [7]. The *F. gigantica* eggs were operculated and yellow–brown stained, while the Paramphistomidae eggs were somewhat larger and not stained yellow. The number of trematode eggs was determined by thoroughly examining every section of the Petri dish. To ensure accuracy, two lab technicians independently examined the same samples. The parasite eggs were counted and identified based on specific morphological features [37]. To calculate the eggs per gram of feces, the number of specific parasite eggs present in a 10 g sample was divided by ten.

### 2.4. Snail Collection and Examination

Snail intermediate hosts of trematode parasites were collected from selected water bodies in each study village using scoops and by hand picking directly from freshwater vegetation for 15–30 min at each sampling [38]. At each sampling site, the GPS coordinates were recorded using GPS devises (Garmin, Olathe, KS, USA). Snails were identified morphologically using the field guide to African freshwater snails [11,39]. Further, each of the snails was placed in a 10 mL beaker filled with 6 mL of distilled water and exposed to light overnight to stimulate cercariae shedding. The following morning, the water in each beaker was poured into Petri dishes and examined under the x40 objective stereoscopic microscope to examine for the presence of cercariae. The harvested cercariae were identified morphologically using published keys [40].

### 2.5. Mapping of Study Villages and Study Farms

The study villages and farms participating in this study were geographically mapped using handheld differential geographic global positioning system (GPS) units (Trimble Navigation Ltd., Sunnyvale, CA, USA), which had an approximate accuracy of ±1 m. The GPS data were downloaded with differential correction into a GPS database (GPS Pathfinder Office 2.8, Trimble Navigation Ltd., Sunnyvale, CA, USA) and the mapping process was carried out using ArcView version 9.2 software (Environmental Systems Research Institute Inc., Redlands, CA, USA)

### 2.6. Statistical Analysis

All the information collected from the field and coprological laboratory analyses was entered into the Census and Survey Processing System (CSPro 8.0.0) software (U.S. Census Bureau, Suitland, MD, USA). The cleaned dataset was transferred to Stata version 15.1 (Stata Corporation, College Station, TX, USA). The data were summarized as descriptive statistics and the proportions were used to summarize the categorical variables, which were then compared using the chi-square test. The egg count data (EPGs) underwent log transformation [log10(EPG+1)], and differences in the EPGs between sex, age, and breed were examined using the two-sample t-test, while the one-way analysis of variance (ANOVA) was used to compare the EPGs between different body conditions. Being positive for at least one parasite egg was the criterion for considering an animal to be infected. Statistical significance was considered at *p* values ≤ 0.05. The associations between parasite infections and risk factors were tested using multiple logistic regression analysis. The QGIS^®^ spatial software version 2.2 was used to represent the distribution of trematode infections by village and district.

## 3. Results

### 3.1. Animal Population Characteristics

A total of 1367 domesticated ruminants (739 cattle, 319 goats, and 309 sheep) were sampled and examined for trematode infections. Most of the sampled animals were female, 57.9% for cattle and 64.4% for sheep. However, the majority of sampled goats, at 83.7%, were males. In terms of age, adult animals constituted the majority of the animals sampled, 73.5% for cattle, 61.1% for goats, and 63.1% for sheep.

### 3.2. Prevalence of F. gigantica, Paramphistomidae and S. bovis

The overall prevalence of trematode infections was 65.7%. The individual frequency of *F. gigantica*, Paramphistomidae and *S. bovis* (based on egg morphology, Figure 2) was 35.1%, 60.2% and 3.1%, respectively (Table 1). All the animal species (cattle, goats and sheep) were co-infected with all three parasite species, but sheep were more co-infected than the other animal species (Table 1). The most frequently observed co-infections were between *F. gigantica* and Paramphistomidae at 29.7% (95% CI: 27.3–32.2%).

Bariadi district (Simiyu region) was found to have a high burden of *F. gigantica* (43%, 95% CI: 36.2–50.0), Paramphistomidae (84.1%, 95% CI: 78.3–88.8) and *S. bovis* infection (7.7%, 95% CI: 4.4–12.2) in cattle compared to Misungwi (Mwanza region) and Iringa rural district (Iringa region). Comparatively, the Simiyu region also exhibited the highest prevalence of *F. gigantica*, Paramphistomidae and *S. bovis* in cattle, goats and sheep compared to the other study regions.

The associations between the trematode infections and the demographic characteristics (sex, age, and breed) of cattle, goats, and sheep are shown in Table 2, Table 3, and Table 4, respectively. Adult cattle (24+ months) were at a higher risk of infection with Paramphistomidae (AOR: 1.98; 95% CI: 1.40–2.78) and *S. bovis* (AOR: 8.5; 95% CI: 1.12–64.19) than younger cattle (Table 3). Locally bred cattle were more susceptible to Paramphistomidae infection (AOR: 8.19; 95% CI: 1.76–38.09) than crossbred cattle and at low risk of infection with *F. gigantica* (AOR: 0.61; 95% CI: 0.19–1.92) and *S. bovis* (AOR: 0.25; 95% CI: 0.03–2.21) (Table 2).

Adult goats (24+ months) were more susceptible to *F. gigantica* infection (AOR: 1.59; 95% CI: 0.92–2.77), Paramphistomidae (AOR: 1.07; 95% CI: 0.8–1.68) and *S. bovis* (AOR: 3.9; 95% CI: 0.46–32.82) than younger goats; however, these associations were not statistically significant (Table 3). Compared with male goats, female goats were less likely to be infected with *F. gigantica* (AOR: 0.83; 95% CI: 0.47–1.46) and *S. bovis* (AOR: 0.97; 95% CI: 0.18–5.12) (Table 3).

Female sheep had a 1.63 times higher likelihood of being infected with *S. bovis* compared to male sheep (AOR: 1.63; 95% CI: 0.43–6.17) (Table 4). Adult sheep had higher odds of being infected with *F. gigantica* (AOR: 2.11; 95% CI: 1.28–3.51), Paramphistomidae (AOR: 1.93; 95% CI: 1.20–3.10), and *S. bovis* (AOR: 2.94; 95% CI: 0.63–13.71) compared to younger sheep. The differences in the infection rates between age groups were statistically significant for *F. gigantica* (*p* < 0.004) and Paramphistomidae (*p* < 0.007), but not for *S. bovis* (Table 4).

### 3.3. Trematode Infection in Snail Intermediate Hosts

A total of 5126 snails intermediate hosts were collected, comprising 1000 (19.5%) *Lymnaea (Radix) natalensis*, 244 (4.7%) *Biomphalaria* species, 1082 (21.1%) *Bulinus tropicus,* 2036 (39.7) *Bulinus nasutus* and 764 (14.9) *Bulinus globosus*. The majority of the snails were collected in ponds. A total of 75 *L. (radix) natalensis* (7.5%) were found to shed *Fasciola* cercariae, while 125 (3.2%), *Bulinus* snails were found to shed schistosome cercariae, but *Biomphalaria* snails did not shed any schistosome cercariae.

### 3.4. Spatial Distribution of Infections with F. gigantica, Paramphistomidae and S. bovis

Infections with *F. gigantica*, Paramphistomidae and *S. bovis* showed varying prevalence across regions and districts, with the Lake Victoria zone exhibiting the highest prevalence (Figure 3 and Table 5). Pugu village in the Simiyu region had the highest prevalence rates of *S. bovis* in cattle, goats, and sheep, with rates of 7.8% (95% CI: 3.4–14.9%), 7.4% (95% CI: 2.0–17.9%), and 12.7% (95% CI: 5.3–24.5%), respectively.

*F. gigantica* was found to be more prevalent in Kanyelele village, Misungwi district (Mwanza region) in sheep, at a prevalence of 70.0% (95% CI: 50.6–85.2), while for cattle, the highest prevalence was observed in Usengelindete village (Iringa region), at 54.6% (95% CI: 44.8–64.2), and for goats, in Itubukilo village (Simiyu region), at 42.1% (95% CI: 29.1–55.0).

The highest prevalence of Paramphistomidae infections was found in Itubukilo A Village for both cattle and sheep, at 88.6% (95% CI: 80.9–93.9%) and 88.7% (95% CI: 77.0–95.7%), respectively. The highest prevalence of Paramphistomum infections was observed in Lupembelwasenga village (80.0%, 95% CI: 51.9–95.7%).

### 3.5. Fecal Egg Counts

The means (±SE) of the *Fasciola*, Paramphistomidae and *Schistosoma* egg counts per gram of feces in cattle were 2.4 (±) 0.06, 3.0 (±) 0.06, and 0.8 (±) 0.05, respectively, with ranges of 0.7 to 6.1, 0.7 to 6.3, and 0.7 to 1.4. In goats, the mean EPG were 2.5 (0.14), 2.6 (0.09), and 0.8 (0.10), with ranges of 0.7 to 5.9, 0 to 6.2, and 0.7 to 1.4. For sheep, the averages were 2.6 (0.11), 2.9 (0.10), and 0.9 (0.09), with ranges of 0.7 to 6.6, 0.7 to 6.5, and 0.7 to 1.4, respectively.

In goats, there was a significant correlation between age and the average EPG for *Fasciola* (*p* < 0.011). For Paramphistomidae infections, the mean egg count showed a significant correlation with the body condition (*p* < 0.028). Paramphistomum infections had a higher mean fecal egg count per gram of feces than *Fasciola* and *Schistosoma* infections.

## 4. Discussion

The burden of trematode infections of *F. gigantica*, Paramphistomidae and *S. bovis* in domesticated ruminants in various ecological zones (the Lake Victoria zone (Mwanza and Simiyu regions) and Southern Highlands (Iringa region)) of Tanzania has been investigated in this study.

Cattle showed a higher prevalence of *S. bovis*, Paramphistomidae and *F. gigantica* infections compared to small ruminants, which is consistent with the findings from the study conducted in Côte d’Ivoire by Kouadio et al. [9]. This could be attributed to the fact that goats primarily consume leaves and heaths in elevated areas, while sheep graze on open land, and cattle graze near water bodies. As a result, cattle are at a greater risk of being exposed to water and vegetation where the infective parasite stage is present, leading to infection.

This study found that Paramphistomidae infections had the highest mean fecal egg count per gram of feces (EPG), followed by *F. gigantica* and *S. bovis*. A similar pattern of infection load for these three trematodes has been documented in various regions of Tanzania and other parts of Africa [34,41]. The lack of control programs against trematode infections leads to livestock owners medicating their domesticated ruminants with commonly used anthelminthic drugs such as albendazole and nitroxinil without a prescription from a veterinary officer. This may lead to the administration of an incorrect dosage, which will accelerate the emergence of resistant trematodes parasite populations, which in turn contributes to high-infection-intensity infections, which maintains the parasite transmission cycle.

It was found that Paramphistomidae and *F. gigantica* infections are the predominant trematode infections in cattle, goats, and sheep in the studied areas. This aligns well with earlier studies conducted in Tanzania and other parts of Africa [8,34,38]. The high prevalence of Paramphistomidae infections could be attributed to the fact that the adult parasite is highly productive and capable of surviving in the host for extended periods. Also, they have a wider range of snail intermediate hosts compared to *Fasciola* and *Schistosoma* [42]. Paramphistomidae parasites are known for their high reproductive capacity and their ability to thrive in harsh conditions [43]. Additionally, many broad-spectrum anthelmintics, such as albendazole and triclabendazole, which are commonly used to treat nematode and trematode infections, have minimal or no effect on Paramphistomidae infections [44].

In regards to the low prevalence of *S. bovis* infection, the sedimentation technique has been shown to have a low sensitivity depending on the protocol. This may be contributed to by factors such as variation in the distribution of eggs within a single fecal specimen, daily fluctuations in fecal production and consistency in the host and daily fluctuations related to the oviposition patterns of the parasite [45,46,47].

The time samples are left to sediment affects the sensitivity because of the rapid hatching of *Schistosoma* eggs, which may occur during the sedimentation process before the sediment is observed under the microscope [48]. In fact, after exposure to water, *Schistosoma* eggs can hatch within 20 min [49]. In this study, the total time the eggs were left to sediment in water was 45 min. It may also be because not all the *Schistosoma* eggs are excreted in the feces, as many are left trapped in tissue [50]. Lastly, the host immune response against Schistosome parasites is directed toward the suppression of worm fecundity, resulting in reduced egg output rather than elimination of adult worms [51].

The higher prevalence of *F. gigantica* could be attributed to the higher transmission of *Fasciola* parasites in the study regions, which in turn could be attributed to the suitable environment of both the parasite and the snail intermediate host. In addition, fecal examination appears to be more sensitive for *Fasciola* egg detection. A study conducted in Southern Ethiopia also demonstrated a higher prevalence of *F. gigantica* in cattle [52]. However, the low prevalence of *F. gigantica* in goats observed in the Southern Highlands (Iringa region) was due to the small number of goats sampled in the area compared to the number of cattle and sheep. The predominant co-infection of trematodes in this study was observed between *F. gigantica* and Paramphistomidae parasites. This could be attributed to the similarities in the life cycles of these parasites, which both rely on lymnaeid snails as intermediate hosts [53,54].

The observed distribution of trematode infections in this study is likely determined by the natural ecology of the study areas. The presence of water bodies such as rivers, ponds, and streams not only provide habitats for snail intermediate snail hosts of the parasites [19] but also animal–water contact sites (for schistosomiasis) and aquatic plants that can be consumed by domesticated ruminants (fascioliasis). The observed snail intermediate hosts in this study include snails from the genus *Lymnaea*, *Bulinus* and *Biomphalaria*. Similar snail intermediate hosts have been reported in a previous study in the Southern Highlands of Tanzania [19], which play essential roles in maintaining the transmission cycle of trematode infections.

In addition, the livestock management system plays an essential role in maintaining the transmission of schistosomiasis and other trematode infections of domestic ruminants. In traditional livestock management systems, domesticated ruminants graze near and drink water from rivers and ponds, depositing feces containing the parasite eggs and hence perpetuating the transmission cycle.

The results of previous studies conducted in the Southern Highlands of Tanzania indicated that transmission of trematode infections is impacted by the management systems of domesticated ruminants. Indeed, a study revealed that in areas where traditional livestock management systems were practiced, the prevalence of trematodes infections was high. The prevalence was moderate in large-scale dairy management systems and lowest in small-scale dairy management systems [11]. This finding aligns with a study conducted in Mali, which demonstrated that factors such as the climate, the presence of water bodies, and the type of domesticated ruminant rearing systems significantly affected the prevalence of trematode infections [55].

Adult cattle were more infected by Paramphistomidae and *S. bovis* infections than younger animals, likely due to prolonged exposure to contaminated water while grazing. This infection pattern has also been observed in Nigeria [56] and in Tanzania [19]. Further, it could also be possible that as domesticated ruminants become older, their immunity against trematode infections decreases and hence they experience higher infection rates [57]. On the contrary, a study found a higher prevalence rate of fascioliasis in weaners than in adults [58] and hypothesized that older animals developed acquired immunity that results in increased resistance [12].

The study limitations include the small sample size for goats and sheep in comparison to cattle across all the study villages/regions, which means the results should be interpreted with caution. Additionally, this study did not document the treatment history, which could have impacted the prevalence of trematode infections. The identification of parasite species relied solely on the egg morphology. Nonetheless, this study underscores the significance of trematode infections of veterinary and public health importance in Tanzania. It is also the first study to provide a comprehensive overview of the magnitude and distribution of these trematode infections among domesticated ruminants in the Lake Victoria zone (Mwanza and Simiyu regions) of Tanzania.

## 5. Conclusions

This study revealed that Paramphistomidae and *F. gigantica* infections were the more prevalent trematode infections in cattle, goats, and sheep in the Lake Victoria zone (Mwanza and Simiyu) and the Southern Highlands (Iringa region) of Tanzania. Based on these findings, it is crucial to implement effective trematode and snail intermediate host control strategies to reduce infections and minimize economic loses. However, the strategic use of anthelmintics effective against mixed trematode infections and improvements in livestock management practices, such as keeping domesticated ruminants away from grazing areas that are heavily contaminated by trematode eggs, are highly recommended.

## Figures and Tables

**Figure 1 vetsci-11-00595-f001:**
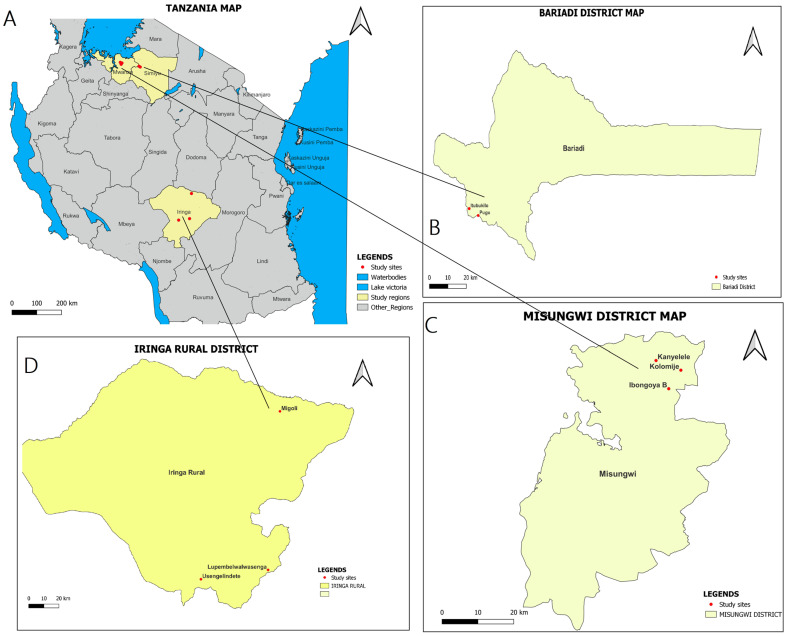
A map showing the study areas: (**A**) Tanzania administrative map, (**B**) Bariadi District Council, (**C**) Misungwi District Council and (**D**) Iringa Rural District Council.

**Figure 2 vetsci-11-00595-f002:**
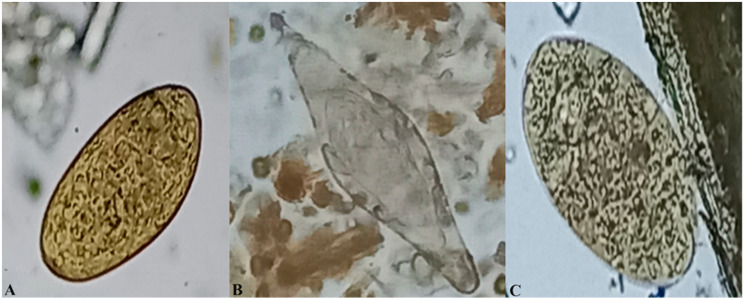
*Fasciola* egg (**A**), *S. bovis* egg (**B**) and Paramphistomidae egg (**C**).

**Figure 3 vetsci-11-00595-f003:**
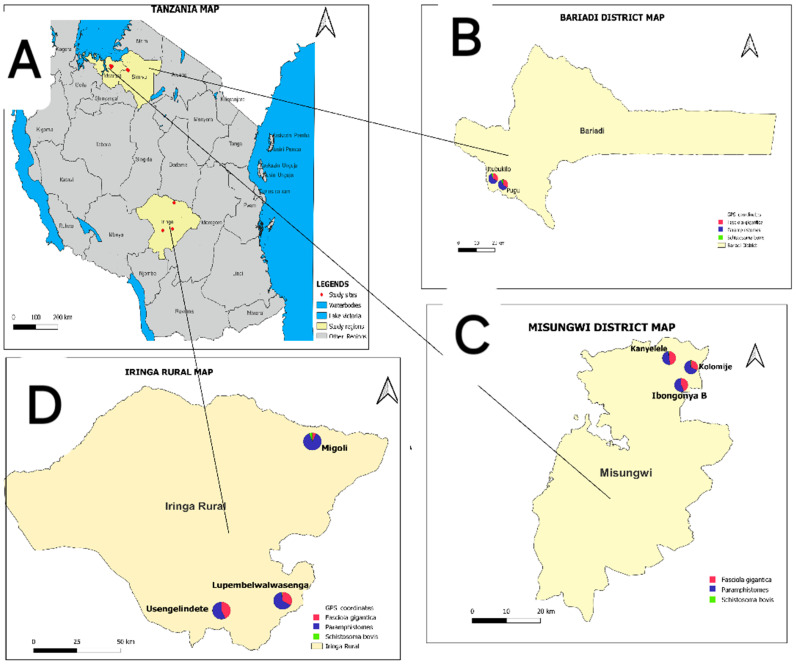
Administrative map of Tanzania showing the distribution of trematode infections in the study villages in Misungwi, Bariadi and Iringa rural districts, Tanzania. (**A**) Tanzania administrative map, (**B**) Bariadi District Council, (**C**) Misungwi District Council and (**D**) Iringa Rural District Council.

**Table 1 vetsci-11-00595-t001:** The prevalence of trematode infections (*F. gigantica*, Paramphistomidae and *S. bovis*) by animal species.

	Cattle N = 739		Goats N = 319		Sheep N = 309		Total N = 1367	
	**++**	**% (95% CI)**	**++**	**% (95% CI)**	**++**	**% (95% CI)**	**++**	**% (95% CI)**
*F. gigantica*	289	39.1 (35.6–42.7)	75	23.5 (19.0–28.6)	116	37.5 (32.1–43.2)	480	35.1 (32.6–37.7)
Paramphistomidae	484	65.5 (61.9–68.9)	155	48.6 (43.0–54.2)	184	59.6 (53.8–65.1)	823	60.2 (57.6–62.8)
*S. bovis*	24	3.3 (2.1–4.8)	7	2.2 (0.9–4.5)	12	3.9 (2.0–6.7)	43	3.1 (2.3–4.2)
*F. gigantica* + Paramphistomidae	247	33.4 (30.0–37.0)	65	20.4 (16.1–25.2)	94	30.4 (25.3–35.9)	406	29.7 (27.3–32.2)
*F. gigantica* + *S. bovis*	16	2.2 (1.2–3.5)	5	1.6 (0.5–3.6)	11	3.6 (1.8–6.3)	32	2.3 (1.6–3.3)
Paramphistomidae + *S. bovis*	23	3.1 (1.9–4.6)	7	2.2 (0.9–4.5)	11	3.6 (1.8–6.3)	41	3.0 (2.2–4.0)
*F. gigantica* + Paramphistomidae *+ S. bovis*	16	2.2 (1.2–3.5)	5	1.6 (0.5–3.6)	10	3.2 (1.6–5.9)	31	2.3 (1.5–3.2)
Overall Prevalence	527	71.3 (67.9–74.5)	165	51.7 (46.1–57.3)	206	66.7 (61.1–71.9)	898	65.7 (63.1–68.2)

N: examined domesticated ruminants, ++: infected domesticated ruminants, CI: confidence interval.

**Table 2 vetsci-11-00595-t002:** Multiple logistic regression analysis of variables associated with *F. gigantica*, Paramphistomidae and *S. bovis* infections among cattle.

		*F. gigantica*					Paramphistomidae					*S. bovis*				
	N	++	%	AOR	95% CI	*p*-Value	++	%	AOR	95% CI	*p*-Value	++	%	AOR	95% CI	*p*-Value
Sex																
Male	311	114	36.7				207	66.6				6	1.93			
Female	428	175	40.9	1.17	0.87–1.59	0.303	277	64.7	0.86	0.63–1.19	0.368	18	4.21	1.93	0.75–4.98	0.171
Age																
6–24 months	195	71	36.4				104	53.3				1	0.51			
24+ months	543	218	40.2	1.16	0.83–1.64	0.380	379	69.8	1.98 *	1.40–2.78	<0.001	23	4.24	8.5 *	1.12–64.19	0.038
Breed																
Cross	12	6	50				2	16.7				1	8.33			
Local	727	283	38.9	0.61	0.19–1.92	0.395	482	66.3	8.19 *	1.76–38.09	0.007	23	3.16	0.25	0.03–2.21	0.211

N: examined domesticated ruminants, ++: infected domesticated ruminants, CI: confidence interval, AOR: adjusted odds ratio, * *p* < 0.05.

**Table 3 vetsci-11-00595-t003:** Multiple logistic regression analysis of variables associated with *F. gigantica*, Paramphistomidae and *S. bovis* infection among goats.

		*F. gigantica*					Paramphistomidae					*S. bovis*				
	N	++	%	AOR	95% CI	*p*-Value	++	%	AOR	95% CI	*p*-Value	++	%	AOR	95% CI	*p*-Value
Sex																
Male	267	69	25.8				135	50.6				7	2.6			
Female	52	6	11.5	0.83	0.47–1.46	0.512	20	38.5	1.19	0.73–1.95	0.486	0	0.0	0.97	0.18–5.12	0.972
Age																
6–24 months	124	23	18.6				59	47.6				1	0.8			
24+ months	195	52	26.7	1.59	0.92–2.77	0.099	96	49.2	1.07	0.68–1.68	0.764	6	3.1	3.9	0.46–32.82	0.210
Breed																
Cross	0															
Local	319	75	23.5	NA			155	48.6	NA			7	2.2	NA		

N: examined domesticated ruminants, ++: infected domesticated ruminants, CI: confidence interval, AOR: adjusted odds ratio.

**Table 4 vetsci-11-00595-t004:** Multiple logistic regression analysis of variables associated with *F. gigantica*, Paramphistomidae and *S. bovis* infection among sheep.

		*F. gigantica*					Paramphistomidae					*S. bovis*				
	N	++	%	AOR	95% CI	*p*-Value	++	%	AOR	95% CI	*p*-Value	++	%	AOR	95% CI	*p*-Value
Sex																
Male	108	42	38.9				69	63.9				3	2.7			
Female	199	73	36.7	0.89	0.55–1.46	0.650	115	57.8	0.76	0.46–1.24	0.265	9	4.5	1.63	0.43–6.17	0.472
Age																
6–24 months	112	30	26.8				56	50.0				2	1.8			
24+ months	195	85	43.6	2.11 *	1.28–3.51	0.004	128	65.7	1.93 *	1.20–3.10	0.007	10	5.1	2.94	0.63–13.71	0.168
Breed																
Cross	0	0	0				0	0				0	0			
Local	309	116	37.5	NA			184	59.6	NA			12	3.9	NA		

N: examined domesticated ruminants, ++: infected domesticated ruminants, CI: confidence interval, AOR: adjusted odds ratio, * *p* < 0.05.

**Table 5 vetsci-11-00595-t005:** The prevalence of trematode infections (*F. gigantica*, Paramphistomidae, and *S. bovis*) by region and animal species.

	*F. gigantica*						Paramphistomidae						*S. bovis*						Co-Infections
	Cattle		Goat		Sheep		Cattle		Goat		Sheep		Cattle		Goat		Sheep		Domesticated Ruminants
Region	**++**	**% (95% CI)**	**++**	**% (95% CI)**	**++**	**% (95% CI)**	**++**	**% (95% CI)**	**++**	**% (95% CI)**	**++**	**% (95% CI)**	**++**	**% (95% CI)**	**++**	**% (95% CI)**	**++**	**% (95% CI)**	**N (%)**
SIMIYU	89	43 (36.2–50.0)	42	37.8 (28.8–47.5)	52	48.1 (38.4–58.0)	174	84.1 (78.3–88.8)	71	64.0 (54.3–72.9)	81	75 (65.7–82.8)	16	7.7 (4.4–12.2)	5	4.5 (1.5–10.2)	12	11.1 (0.6–18.6)	364 (26.6)
IRINGA	110	35.7 (30.4–41.3)	3	2.6 (0.5–7.4)	23	20.7 (13.6–29.4)	186	60.4 (54.7–65.9)	40	34.8 (26.1–44.2)	57	51.4 (41.7–60.9)	6	2 (0.7–4.2)	1	0.9 (0.0–4.7)	0	0 (0.0–0.3)	235 (17.2)
MWANZA	90	40.2 (33.7–46.9)	30	32.3 (22.9–42.7)	41	45.6 (35.0–56.4)	124	55.4 (48.6–62.0)	44	47.3 (36.9–57.9)	46	51.1 (40.3–61.8)	2	0.9 (0.1–3.2)	1	1.1 (0.0–5.8)	0	0 (0.0–0.4)	224 (16.4)

N (%): domesticated ruminants with co-infections, ++: infected domesticated ruminants, CI: confidence interval, Co-infections: mixed infections with at least two parasites.

## Data Availability

The data presented in this study are available on request from the corresponding author.

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
