# Peer review of "Prevalence, Distribution and Risk Factors for Trematode Infections in Domesticated Ruminants in the Lake Victoria and Southern Highland Ecological Zones of Tanzania: A Cross-Sectional Study"

_vetsci, 2024, doi:10.3390/vetsci11120595_

Round 1

Reviewer 1 Report

Comments and Suggestions for Authors

General comments

The manuscript shows several shortcomings and should be improved.

Throughout the manuscript, including the References and Keywords sections Tables and Figures, genera and species of parasite names should be written in Italics. Moreover, the genus should be written with the first letter in capital. Furthermore, the genus of a parasite species name should be written in full the first time it is mentioned in the Abstract and in the body of the manuscript. Finally, the word “paramphistomes” should be replaced with Paramphistomidae (or rumen flukes).

In the abstract, “AOR” and “CI” should be written in full the first time it appears in the text.

Introduction

-At line 76 it should be better to replace the word “parts” with “areas or regions”-

Materials and Methods

-(lines 127-129) Authors should detail the methods used to distinguish Fasciola spp. and Paramphistomid eggs considering their statement: “Eventually, all the sediment was placed into a Petri dish, stained with 1% methylene blue, and observed under a microscope at 10x magnification for parasitic eggs”.

-(lines 129-133) “The number of trematode eggs was determined by thoroughly examining every section of the Petri dish. To ensure accuracy, two lab technicians independently examined the same samples. The eggs were counted and identified based on specific morphological characteristics [35]. To calculate the egg per gram of feces, the number of specific parasite eggs present in a 10 g sample was divided by ten.”. Authors should consider that counting eggs per gram of feces is not a reliable finding in the case of liver flukes. Therefore, obtained data should be discussed accordingly.

- (lines 137) “from Trimble Navigation Ltd. in California, USA”, this is the manufacturer of the instrument and should be inserted in round brackets.

Results

The number of Tables is really excessive and should be significantly reduced. For example, Tables 1 and 2 should be merged into a single table; the main data related to the Multiple logistic regression analysis reported in Tables 3, 4 and 5 can be inserted in the text of the results. Similarly, the main data of Tables 6 and 7 can also be inserted in the text of the Results section (especially considering the severe limitations in the meaning of these data).

Discussion

The discussion is poor and should be improved. Moreover, the discussion section cannot be simply a photograph of obtained results (i.e., a repetition of results) with comparisons with previous data (also very few), but also a discussion of results based on previous knowledge about the epidemiology of these trematode infections. For example, it is well known that the prevalence of  Fasciola hepatica and Paramphistomidae is generally higher in adult animals.

- the Authors should explain why in all other sections of the manuscript they speak about Fasciola spp. while in the Discussion they speak about Fasciola gigantica.  Moreover, authors should consider that both Fasciola hepatica and Fasciola gigantica have been reported in Tanzania (see for example, Walker SM. et al., The distribution of Fasciola hepatica and Fasciola gigantica within southern Tanzania--constraints associated with the intermediate host. Parasitology. 2008 Apr;135(4):495-503. doi: 10.1017/S0031182007004076). Therefore, authors should detail in Materials and Methods the methods used to differentiate these two species and, also, specify the methods used to identify all parasites reported in the manuscript (and the related references used).

-(lines 23-25) Ivermectin has no efficacy against trematodes! This sentence should be amended accordingly.

Conclusions

Useful chemical and non-chemical methods for the control of these infections should be proposed by the authors for the investigated areas.

References

In addition to the changes requested above, authors should also check the spacing in references.

Comments on the Quality of English Language

The English is fine, and only minor editing are required.

Author Response

Response to Reviewer 1 Comments

1. Summary

Thank you very much for taking the time to review this manuscript. Please find the detailed responses below and the corresponding revisions/corrections highlighted in the re-submitted files.

pen Review

Quality of English Language

( ) I am not qualified to assess the quality of English in this paper.
( ) The English is very difficult to understand/incomprehensible.
( ) Extensive editing of English language required.
( ) Moderate editing of English language required.
(x) Minor editing of English language required.
( ) English language fine. No issues detected.

Yes

Can be improved

Must be improved

Response

Does the introduction provide sufficient background and include all relevant references?

( )

(x)

( )

( ) Improved

Is the research design appropriate?

( )

( )

(x)

( ) Improved

Are the methods adequately described?

( )

( )

(x)

( ) Improved

Are the results clearly presented?

( )

( )

(x)

( ) Improved

Are the conclusions supported by the results?

( )

( )

(x)

( ) Modified

  1. General comments

Comments 1: [Throughout the manuscript, including the References and Keywords sections Tables and Figures, genera and species of parasite names should be written in Italics. Moreover, the genus should be written with the first letter in capital. Furthermore, the genus of a parasite species name should be written in full the first time it is mentioned in the Abstract and in the body of the manuscript. Finally, the word “paramphistomes” should be replaced with Paramphistomidae (or rumen flukes).]

Response 1: Thank you for pointing this out. We agree with this comment. Therefore, we have revised throughout the manuscript.

 In the abstract, “AOR” and “CI” should be written in full the first time it appears in the text.

Response 1: Thank you for pointing this out. we have corrected.

3. Point-by-point response to Comments and Suggestions for Authors

Point 1: [-At line 76 it should be better to replace the word “parts” with “areas or regions”-

Response 1: Agree. We have, changed.

Point 2: Materials and Methods -(lines 127-129) Authors should detail the methods used to distinguish Fasciola spp. and Paramphistomid eggs considering their statement: “Eventually, all the sediment was placed into a Petri dish, stained with 1% methylene blue, and observed under a microscope at 10x magnification for parasitic eggs”.

Response 2: We agree with this comment. We have revised the manuscript; these changes can be found in line 151-154.

Point 3: [-(lines 129-133) “The number of trematode eggs was determined by thoroughly examining every section of the Petri dish. To ensure accuracy, two lab technicians independently examined the same samples. The eggs were counted and identified based on specific morphological characteristics [35]. To calculate the egg per gram of feces, the number of specific parasite eggs present in a 10 g sample was divided by ten.”. Authors should consider that counting eggs per gram of feces is not a reliable finding in the case of liver flukes. Therefore, obtained data should be discussed accordingly.]

Response 3: The findings has been discussed accordingly

Point 4: [ (lines 137) “from Trimble Navigation Ltd. in California, USA”, this is the manufacturer of the instrument and should be inserted in round brackets.]

 Response 4: Corrected: Changes found in line 173.

Point 5: [ Results, the number of Tables is really excessive and should be significantly reduced. For example, Tables 1 and 2 should be merged into a single table; the main data related to the Multiple logistic regression analysis reported in Tables 3, 4 and 5 can be inserted in the text of the results. Similarly, the main data of Tables 6 and 7 can also be inserted in the text of the Results section (especially considering the severe limitations in the meaning of these data).]

Response 5: Thank you for pointing this out. We agree with this comment, we have reduced the number of tables from 7 to 4, however table 1 and 2 cannot be merged because table 1 shows prevalence by species and tables 2 shows prevalence by regions. Instead, we have removed table 2 which is replaced by figure 4 under “Spatial distribution of infections with F. gigantica, Paramphistomidae and S. bovis sub-section”. Tables 3,4,5, now will read table 2,3,4,. Finally, the main data of Tables 6 and 7 has been inserted in the text of the results section.

Discussion

Point 6: [The discussion is poor and should be improved. Moreover, the discussion section cannot be simply a photograph of obtained results (i.e., a repetition of results) with comparisons with previous data (also very few), but also a discussion of results based on previous knowledge about the epidemiology of these trematode infections. For example, it is well known that the prevalence of Fasciola hepatica and Paramphistomidae is generally higher in adult animals.]

Response 6[The discussion section has been revised accordingly}

-

Point 7: [the Authors should explain why in all other sections of the manuscript they speak about Fasciola spp. while in the Discussion they speak about Fasciola gigantica.  Moreover, authors should consider that both Fasciola hepatica and Fasciola gigantica have been reported in Tanzania (see for example, Walker SM. et al., The distribution of Fasciola hepatica and Fasciola gigantica within southern Tanzania--constraints associated with the intermediate host. Parasitology. 2008 Apr;135(4):495-503. doi: 10.1017/S0031182007004076).

 Therefore, authors should detail in Materials and Methods the methods used to differentiate these two species and, also, specify the methods used to identify all parasites reported in the manuscript (and the related references used).]

Response 7: [Agreed. both Fasciola hepatica and Fasciola gigantica have been reported in Tanzania (see for example, Walker SM. et al.  with F.hepatica reported in Kitulo plateau at attitude above 1500M. We have revised our manuscript in Materials and Methods and result section by incorporating snail data, to emphasize this point, the animal and snail sampling were conducted in areas with altitude below 1500m from the sea level of which the collected snails was Radix natalensis the intermediate host of F.gigantica

-(lines 23-25) Ivermectin has no efficacy against trematodes! This sentence should be amended

accordingly.

Response: The sentence has been amended

Conclusions

Useful chemical and non-chemical methods for the control of these infections should be proposed by the authors for the investigated areas.

Response: Agreed

References

In addition to the changes requested above, authors should also check the spacing in references.

Response: Corrected

Reviewer 2 Report

Comments and Suggestions for Authors

span lang="EN-US"This is a very interesting paper concerning on /spanspan lang="EN-US"prevalence, distribution, and risk factors for trematode infections in domesticated ruminants in two ecological zones of Tanzania. Prevalence of trematode infections was found to be 65.7%. The individual frequency of F. gigantica, paramphistomes, and Sbovis  was 35.1%, 60.2% and 3.1%, respectively. Adult cattle were more likely to be infected with paramphistomes (AOR: 1.98, 95% C: 1.40-2.78) and S. bovis (AOR:8.5: 95% CI: 1.12--64.19) than weaners. It was found that trematode infections in domesticated ruminants are prevalent across Tanzania, therefore effective and community-acceptable prevention and control strategies are highly needed. It is certainly an interesting topic, be worth for publication. Thus I recommend “accept with minor revisio” to this manuscript./span/p p class="MsoNormal"span lang="EN-US" /span/p p class="MsoNormal"span lang="EN-US" /span/p p class="MsoNormal" style="margin-bottom: 8pt" align="left"span lang="EN-US"Minor issues/span/p p class="MsoListParagraph" style="margin: 0cm 0cm 8pt 18pt; text-indent: 0pt" align="left"1.     Now that molecular biology studies are commonplace, why didn’t do molecular identification studies?

2.     Please note the highlighted yellow marker.

Author Response

Article

Prevalence, distribution and risk factors for trematode infections in domesticated ruminants in the Lake Victoria and Southern Highland ecological Zones of Tanzania: A cross-sectional study

Godlisten Shedrack Materu1, 2, 3*, Jahashi Nzalawahe1, Mita Eva Sengupta4, Anna-Sofie Stensgaard4, Abdul Katakweba5, Birgitte J. Vennervald4 and Safari Kinung’hi3

  1. Department of Veterinary Microbiology, Parasitology and Biotechnology, College of Veterinary Medicine and Biomedical Sciences, Sokoine University of Agriculture, P. O. Box 3019, Morogoro, Tanzania; nzalawahej@sua.ac.tz (J.N)
  2. National Institute for Medical Research (NIMR), P. O. Box 538 Tukuyu Research Centre, Mbeya, Tanzania; materu.godlisten@yahoo.com (G.S.M)
  3. National Institute for Medical Research (NIMR), P.O. Box 1462 Mwanza Research Centre, Mwanza, Tanzania; skinunghi@gmail.com (S.K)
  4. Department of Veterinary and Animal Sciences, University of Copenhagen, DYRLAEGEVEJ100.2, DK-1870 FREDERIKSBERG C, Denmark; msen@sund.ku.dk (M.E.S); bjv@sund.ku.dk (B.J.V); asstensgaard@sund.ku.dk (A.S)
  5. Institute of Pest Management, Sokoine University of Agriculture, P. O. Box 3110, Chuo Kikuu, Morogoro, Tanzania; katakweba@sua.ac.tz (A.K)

*  Correspondence: Godlisten Shedrack Materu, materu.godlisten@yahoo.com

Simple Summary: Trematode infections (fascioliasis and schistosomiasis) are a worldwide emerging snail-borne zoonotic disease with a great capacity for expansion due to animal and human movements, climate change, and anthropogenic modifications of freshwater environments. There is a lack of information regarding the prevalence, distribution, and risk factors of trematode infections in domesticated ruminants in Tanzania. This study aims to fill the knowledge gaps by determining the prevalence, distribution, and risk factors for trematode infections in domesticated ruminants in two ecological zones of Tanzania. Fecal samples were collected from domesticated ruminants and examined for trematode infections using the sedimentation technique. The highest trematode infections Fasciola gigantica (F.gigantica), Paramphistomidae, and Schistosoma bovis (S. bovis)were observed in the Lake Victoria zone (Simiyu region) of Tanzania compared to the Southern Highland zone (Iringa region) of Tanzania. To our knowledge, this is the first study reporting trematode infection in domesticated ruminants in the Lake Victoria and Southern highlands ecological zone of Tanzania. Effective and community-directed prevention and control strategies are highly needed to address trematode infections of domesticated ruminants in these areas.

Abstract: Trematode infections cause long-term suffering and debilitation, posing a significant threat to global animal health and production and leading to considerable economic losses. Studies on the epidemiology and control of these infections in Tanzania are limited. The few available studies have been conducted in abattoir settings. The study aimed to fill this knowledge gap by determining the prevalence, distribution, and risk factors of trematode infections in domesticated ruminants in two different ecological zones of Tanzania. A cross-sectional study was conducted in Lake Victoria and the Southern highlands ecological zones of Tanzania. Rectal fecal samples were collected and examined for F. gigantica, Paramphistomidae, and S. bovis infections using the sedimentation technique. A total of 1367 domesticated ruminants were sampled and examined for trematode infections. The overall prevalence of trematode infections was found to be 65.7%. The individual prevalence of F. gigantica, Paramphistomidae, and S. bovis (based on egg morphology only) was 35.1%, 60.2%, and 3.1%, respectively. Adult cattle were more likely to be infected with Paramphistomidae (Adjusted odds ratio, (AOR): 1.98; 95%Confidence interval, (CI): 1.40-2.78) and S. bovis (AOR: 8.5; 95% CI: 1.12–64.19) than weaners. It was observed that trematode infections of domesticated ruminants are prevalent in the two ecological zones of Tanzania, therefore effective, community-directed prevention and control strategies are highly needed to address trematode infections of domesticated ruminants in these areas.

Keywords: Domesticated ruminants; trematode infection, ecological zone, Tanzania

  1. Introduction

Trematode infections, particularly fascioliasis and schistosomiasis, are among the most common helminth infections in domestic ruminants worldwide [1, 2]. Adult trematodes are sometimes referred to as "flukes" and the families that contain parasites of veterinary importance include Schistosomatidae, Fasciolidae, Paramphistomatidae, and Dicrocoeliidae [4,5]. The genera Fasciola (liver fluke), Paramphistomidae (rumen/stomach fluke), and Schistosoma (blood fluke) [4] have similar life cycles in which domestic ruminants and humans (except rumen flukes) are definitive hosts, while wild animals serve as reservoirs [5].

The presence of these parasites is widespread in tropical and subtropical regions globally, thriving in areas with favorable climatic, ecological, and hygienic conditions [6,7]. In most African countries south of the Sahara, these infections are considered to be endemic [8]. The presence and distribution of freshwater snail intermediate hosts often influence the occurrence of trematode infections [9, 10, 11].

Fascioliasis is an economically important disease of domestic ruminants, in particular cattle and sheep, and occasionally man. Bovine fascioliasis is regarded as the most important zoonotic helminthic infection in tropical countries with the prevalence ranging from 30% to 90% [12]. The two most important species of Fasciola parasite that are commonly considered the causative agents of fasciolosis are F. hepatica and F. gigantica [3]. The two species F. gigantica [13] and F. hepatica [14] have been widely reported in Tanzania.

There are several genera of paramphistomidae: Paramphistomum, Cotylophoron, Calicophoron, Bothriophoron, Orthocoelium, and Gigantocotyle, of which Paramphistomum is the most common and widespread in ruminants [15]. There are several reports of coproscopic and abattoir surveys indicating the widespread of Paramphistomidae species and Dicrocoelium hospes in Tanzania [16,17,18]. Paramphistomidae are traditionally regarded as having no clinical significance [19] however, heavy infection with immature flukes, which attach to the lining of the upper part of the small intestine, may cause severe disease which may even result in death.[20].

Schistosomiasis is thought to infect more than 165 million animals in Africa [21]. Schistosoma bovis, S. curassoni, and S. mattheei infect livestock such as cattle, sheep, and goats all across Sub-Saharan Africa [22]. In Tanzania, several reports of coproscopic and abattoir surveys indicate widespread distribution of S. bovis [10, 23,16].

In the southern highlands of Tanzania, about 100% of the total condemnations of bovine liver in slaughter slabs, poor growth rate, reduced milk production, and infertility are attributed to trematode infections [24, 17]. In Tanzania, bovine trematode infections have been reported in all geographical zones by abattoir surveys [25-29]. However, most studies on the epidemiology and control of these parasites in cattle have been carried out in the southern highlands [6,13,14,30]. In these studies, the prevalence of fascioliasis and schistosomiasis ranged from 18% to 94% in domesticated ruminants depending on the production system used [24, 17].

Despite these findings, there is a paucity of information regarding the magnitude, distribution, and risk factors for trematode infections in domesticated ruminants in the Lake Zone and other areas of Tanzania. This study was therefore conducted to address this gap and generate baseline data that will provide the scientific evidence needed to design effective and locally accepted control interventions against trematode infections.

  1. Materials and Methods

2.1.Study areas

The study was conducted from March to May 2023 in the Misungwi, Bariadi, and Iringa rural district Councils of the Mwanza, Simiyu, and Iringa regions, respectively. Within Misungwi district, the study was carried out in Kanyelele, Koromije, and Ibongoya B villages. In Bariadi district, the study was conducted in Pugu and Itubukilo A villages. On the other hand, in Iringa rural district, the study was conducted in Lupembelwasenga, Usengelindeti, and Migori villages (See Figure 1). The study districts and villages were selected purposively based on the available domesticated ruminant population, disease history, and disease ecology described in routine reports from the respective local District Veterinary Offices.

Bariadi and Misungwi district councils are located in the Northwestern part of Tanzania to the Southeast of Lake Victoria. Both districts experience a low rain season from October to December and a high rain season between March and May of each year, while the dry period usually runs from January to the end of February and from June to the end of September of each year. The districts receive 700mm to 950mm of rain annually [31]. The temperatures in the two districts range from 190°C to 290°C [31].

Iringa Rural District Council is located in the Southern highlands of Tanzania. The highland area in the eastern part of the district consists of many hills and valleys, along with numerous permanent rivers, streams, and ponds. In contrast, the flat lowland area on the western side is semi-arid and characterized by dry grazing land with thickets and scattered bushes, as described by Mahoo et al [32]. The annual rainfall in highland areas ranges from 500mm to 2700mm, while in lowland areas; the rainfall is less than 600 mm [33].

Figure 1. A map showing the study areas: A=Tanzania administrative map, B=Bariadi District Council, C=Misungwi District Council and D=Iringa Rural District Council.

2.2.Study design and sampling procedure

The study was a cross-sectional survey whereby rectal fecal samples were collected by trained veterinary assistants from cattle, sheep, and goats randomly selected from household farms in the study villages. Household farm-level information such as household name, sub-village, and flock size was recorded. Individual level animal information such as age (classified as weaners and adults), sex, breed (local-breed and cross-breed), and body condition (lean, median, fat) were also recorded. Unique identification numbers and dates were used to label the samples, after which they were stored in an ice-packed cool box and transported to the laboratory of the National Institute for Medical Research (NIMR), Mwanza Centre, for laboratory examination.

Sample size calculation was performed using the formula: N= (Z2 * p * (1-p))/d2, based on an expected prevalence (p) of bovine fascioliasis of 33% as reported by Nzalawahe et al., [34], and a desired absolute precision (d) of 3% with a 95% confidence interval (CI) as per Thrusfield's [35]. A total of 1367 domesticated ruminants (cattle 739, goats 319 and sheep 309) were sampled in all study districts and examined.

2.3.Coprological examination

Fecal samples from domesticated ruminants were processed using the fecal sedimentation method [36]. In brief, approximately 10 grams of fecal material was placed in a plastic container and mixed with 30ml of tap water. Using 500 ml conical beakers, the fecal suspension was sieved through a wire mesh with an aperture of 250 μm and then filled with tape water. The suspension in a beaker was then allowed to settle for approximately 15 minutes.

Thereafter, the process of decanting the supernatants and re-suspending the sediments in tape water was repeated three times. Eventually, all the sediment was placed into a Petri dish, stained with 1% methylene blue, and observed under light microscope at 10x magnification for examination of parasite eggs [6]. The F. gigantica eggs were operculated and yellow-brown stained while Paramphistomidae eggs were somewhat larger and do not stain yellow. The number of trematode eggs was determined by thoroughly examining every section of the Petri dish. To ensure accuracy, two lab technicians independently examined the same samples. The parasite eggs were counted and identified based on specific morphological feaures [37]. To calculate the egg per gram of feces, the number of specific parasite eggs present in a 10g sample was divided by ten.

2.4. Snail collection and examination

Snail intermediate hosts of trematode parasites were collected from selected water bodies in each study village using scoops and by hand picking directly from freshwater vegetation for 15-30 minutes at each sampling [38]. At each sampling site, GPS coordinates were recorded using GPS devises (Garmin, Olathe, KS). Snails were identified morphologically using the field Guide to African Freshwater Snails [10,39]. Further, each of the snails was placed in a 10ml beaker filled with 6ml of distilled water and exposed to light overnight to stimulate cercariae shedding. The following morning, water in each beaker was poured into Petri dishes and examined under the x40 objective stereoscopic microscope to examine for the presence of cercariae.The harvested cercariae were identified morphologically using published keys [40].

2.5.Mapping of study villages and study farms

The study villages and farms participating in the study were geographically mapped using handheld differential geographic global positioning system (GPS) units (Trimble Navigation Ltd., California, USA), which had an approximate accuracy of ± 1 meter. The GPS data was downloaded with differential correction into a GPS database (GPS Pathfinder Office 2.8, Trimble Navigation Ltd., California, USA) and the mapping process was carried out using ArcView version 9.2 software (Environmental Systems Research Institute Inc., Redlands, CA, USA (Refer to Figure 3 for more details).

2.6.Statistical analysis

All information collected from the field and coprological laboratory analysis were entered into the Census and Survey Processing System (CSPro) software (U.S. Census Bureau, USA). The cleaned dataset was transferred to Stata version 15.1 (Stata Corporation, College Station, TX, USA). The data was summarized as descriptive statistics and proportions were used to summarize categorical variables, which were then compared using the chi-square test. The egg count data (EPGs) underwent log transformation [log10(EPG+1)], and differences in EPGs between sex, age, and breed were examined using the two-sample t-test while the one-way analysis of variance (ANOVA) was used to compare the EPGs between different body conditions. Positive for at least one parasite egg was the criteria for considering an animal to be infected. Statistical significance was considered at P values ≤ 0.05. Associations between parasite infections and risk factors were tested using multiple logistic regression analysis. The QGIS® spatial software version 2.2 was used to represent the distribution of trematode infections by village and district.

  1. Results

3.1. Animal population characteristics

A total of 1367 domesticated ruminants (739 cattle, 319 goats, and 309 sheep) were sampled and examined for trematode infections. Most of the sampled animals were female 57.9% for cattle and 64.4% for sheep. However, the majority of sampled goats at 83.7%were males. In terms of age, adult animals constituted the majority of the animals sampled 73.5% for cattle, 61.1% for goats, and 63.1% for sheep.

3.2. Prevalence of F. gigantica, Paramphistomidae  and S. bovis

The overall prevalence of trematode infections was 65.7%. The individual frequency of F. gigantica, Paramphistomidae and S. bovis (based on egg morphology, Figure 2) was 35.1%, 60.2% and 3.1%, respectively (Table 1). All animal species (cattle, goats and sheep) were co-infected with all three parasite species but sheep were more co-infected than other animal species (Table 1). The most frequently observed co-infections were between F. gigantica and Paramphistomidae at 29.7% (95% CI: 27.3-32.2%).

Bariadi district (Simiyu region) was found to have high burden of F. gigantica (43%, 95% CI: 36.2-50.0), Paramphistomidae (84.1%, 95% CI: 78.3-88.8) and S. bovis infection (7.7%, 95% CI: 4.4-12.2) in cattle compared to the Misungwi (Mwanza region) and Iringa rural district (Iringa region) (Figure 4). Comparatively, the Simiyu region also exhibited the highest prevalence of F. gigantica, Paramphistomidae and S. bovis in goats and sheep compared to the other study regions (Figure 4).

Figure 2. Fasciola egg (A), S. bovis egg (B) and Paramphistomidae egg (C).

Table 1. The prevalence of trematode infections (F.gigantica, Paramphistomidae and S bovis) by animal species

Cattle

N=739

Goats

N=319

Sheep

N=309

Total

N=1367

++

%(95%CI)

++

%(95%CI)

++

%(95%CI)

++

%(95%CI)

F.gigantica

289

39.1 (35.6-42.7)

75

23.5 (19.0-28.6)

116

37.5 (32.1-43.2)

480

35.1 (32.6-37.7)

Paramphistomidae

484

65.5(61.9-68.9)

155

48.6 (43.0-54.2)

184

59.6 (53.8-65.1)

823

60.2 (57.6-62.8)

S.bovis

24

3.3 (2.1-4.8)

7

2.2(0.9-4.5)

12

3.9 (2.0-6.7)

43

3.1(2.3-4.2)

F. gigantica+ Paramphistomidae

247

33.4 (30.0-37.0)

65

20.4(16.1-25.2)

94

30.4(25.3-35.9)

406

29.7(27.3-32.2)

F. gigantica+ S.bovis

16

2.2 (1.2-3.5)

5

1.6(0.5-3.6)

11

3.6 (1.8-6.3)

32

2.3 (1.6-3.3)

Paramphistomidae + S. bovis

23

3.1(1.9-4.6)

7

2.2(0.9-4.5)

11

3.6 (1.8-6.3)

41

3.0 (2.2-4.0)

F. gigantica + Paramphistomidae + S. bovis

16

2.2 ((1.2-3.5)

5

1.6 (0.5-3.6)

10

3.2(1.6-5.9)

31

2.3(1.5-3.2)

Overall Prevalence

527

71.3 (67.9-74.5)

165

51.7(46.1-57.3)

206

66.7(61.1-71.9)

898

65.7(63.1-68.2)

N: Examined domesticated ruminants, ++: infected domesticated ruminants, CI: confidence interval.

The associations between trematode infections and demographic characteristics (sex, age, and breed) of cattle, goats, and sheep are shown in Table 2, Table 3, and Table 4, respectively. Adult cattle (24+ months) were at higher risk of infection with Paramphistomidae (AOR: 1.98; 95% CI: 1.40-2.78) and S. bovis (AOR: 8.5; 95% CI: 1.12–64.19) than younger cattle (Table 3). Locally bred cattle were more susceptible to Paramphistomidae infection (AOR: 8.19; 95% CI: 1.76-38.09) than were crossbred cattle and low risk of infection with F. gigantica (AOR: 0.61; 95% CI: O.19-1.92) and S.bovis (AOR: 0.25; 95% CI: 0.03-2.21) (Table 2).

Adult goats (24+ months) were more susceptible to F. gigantica infection (AOR: 1.59; 95% CI: 0.92-2.77), Paramphistomidae (AOR: 1.07; 95% CI: 0.8-1.68) and S. bovis (AOR: 3.9; 95% CI: 0.46-32.82) than younger goats, however, these associations were not statistically significant (Table 3). Compared with male goats, female goats were less likely to be infected with F. gigantica (AOR: 0.83; 95% CI: 0.47-1.46) and S. bovis (AOR: 0.97; 95% CI: 0.18-5.12) (Table 3).

Female sheep had a 1.63 times higher likelihood of being infected with S. bovis compared to male sheep (AOR: 1.63; 95% CI: 0.43-6.17) (Table 4). Adult sheep had higher odds of being infected with F. gigantica (AOR: 2.11; 95% CI: 1.28-3.51), Paramphistomidae (AOR: 1.93; 95% CI: 1.20-3.10), and S. bovis (AOR: 2.94; 95% CI: 0.63-13.71) compared to younger sheep. The differences in infection rates between age groups were statistically significant for F. gigantica (P < 0.004) and Paramphistomidae (P < 0.007), but not for S. bovis (Table 4).

3.3.Trematode infection in snail intermediate hosts

A total of 5126 snails intermediate hosts were collected comprising 1000 (19.5%) Lymnaea (Radix) natalensis, 244 (4.7%) Biomphalaria species, 1082 (21.1%) Bulinus tropicus, 2036 (39.7) Bulinus nasutus and 764 (14.9) Bulinus globosus). The majority of the snails were collected in ponds. Seventy five L. (radix) natalensis (7.5%) were found to shed Fasciola cercariae while 125 (3.2%), Bulinus snails were found to shed schistosome cercariae but Biomphalaria snails did not shed any schistosome cercariae.

Table 2. Multiple logistic regression analysis of variables associated with F.gigantica,Paramphistomidae and S. bovis infections among cattle.

F.gigantica

Paramphistomidae

S. bovis

N

++

%

AOR

95% CI

p-value

++

%

AOR

95% CI

p- value

++

%

AOR

95% CI

p-value

Sex

Male

311

114

36.7

207

66.6

6

1.93

Female

428

175

40.9

1.17

0.87-1.59

0.303

277

64.7

0.86

0.63-1.19

0.368

18

4.21

1.93

0.75-4.98

0.171

Age

6-24 months

195

71

36.4

104

53.3

1

0.51

24+ months

543

218

40.2

1.16

0.83-1.64

0.380

379

69.8

1.98*

1.40-2.78

<0.001

23

4.24

8.5*

1.12-64.19

0.038

Breed

Cross

12

6

50

2

16.7

1

8.33

Local

727

283

38.9

0.61

0.19-1.92

0.395

482

66.3

8.19*

1.76-38.09

0.007

23

3.16

0.25

0.03-2.21

0.211

N: Examined domesticated ruminants, ++: infected domesticated ruminants, CI: Confidence interval, AOR: Adjusted odds ratio, * p<0.05.

Table 3. Multiple logistic regression analysis of variables associated with F. gigantica, Paramphistomidae and S. bovis infection among goats.

F.gigantica

Paramphistomidae

S. bovis

N

++

%

AOR

95% CI

p-value

++

%

AOR

95% CI

p-value

++

%

AOR

95% CI

p-value

Sex

Male

267

69

25.8

135

50.6

7

2.6

Female

52

6

11.5

0.83

0.47-1.46

0.512

20

38.5

1.19

0.73-1.95

0.486

0

0.0

0.97

0.18-5.12

0.972

Age

6-24 months

124

23

18.6

59

47.6

1

0.8

24+ months

195

52

26.7

1.59

0.92-2.77

0.099

96

49.2

1.07

0.68-1.68

0.764

6

3.1

3.9

0.46-32.82

0.210

Breed

Cross

0

Local

319

75

23.5

NA

155

48.6

NA

7

2.2

NA

N: Examined domesticated ruminants, ++: infected domesticated ruminants, CI: Confidence interval, AOR: Adjusted odds ratio.

Table 4. Multiple logistic regression analysis of variables associated with F. gigantica, Paramphistomidae and S. bovis infection among sheep.

F. gigantica

Paramphistomidae

S. bovis

N

++

%

AOR

95% CI

p-value

++

%

AOR

95% CI

p-value

++

%

AOR

95% CI

p-value

Sex

Male

108

42

38.9

69

63.9

3

2.7

Female

199

73

36.7

0.89

0.55-1.46

0.650

115

57.8

0.76

0.46-1.24

0.265

9

4.5

1.63

0.43-6.17

0.472

Age

6-24 months

112

30

26.8

56

50.0

2

1.8

24+ months

195

85

43.6

2.11*

1.28-3.51

0.004

128

65.7

1.93*

1.20-3.10

0.007

10

5.1

2.94

0.63-13.71

0.168

Breed

Cross

0

0

0

0

0

0

0

Local

309

116

37.5

NA

184

59.6

NA

12

3.9

NA

N: Examined domesticated ruminants, ++: infected domesticated ruminants, CI: Confidence interval, AOR: adjusted odds ratio, * p<0.05.

3.4. Spatial distribution of infections with F.gigantica, Paramphistomidae and S. bovis

Infections with F. gigantica, Paramphistomidae and S. bovis showed varying prevalence across regions and districts, with the Lake Victoria zone exhibiting the highest prevalence (Figure 3 and 4). Pugu village in the Simiyu region had the highest prevalence rates of S. bovis in cattle, goats, and sheep, with rates of 7.8% (95% CI: 3.4-14.9%), 7.4% (95% CI: 2.0-17.9%), and 12.7% (95% CI: 5.3-24.5%) respectively.

  1. gigantica was found to be more prevalent in Kanyelele village, Misungwi district (Mwanza region) in sheep, at a prevalence of 70.0% (95% CI: 50.6-85.2), while for cattle, the highest prevalence was observed in Usengelindete village (Iringa region), at 54.6% (95% CI: 44.8-64.2), and for goats, in Itubukilo village (Simiyu region), at 42.1% (95% CI: 29.1- 55.0).

The highest prevalence of Paramphistomidae infections was found in Itubukilo A Village for both cattle and sheep, at 88.6% (95% CI: 80.9-93.9%) and 88.7% (95% CI: 77.0-95.7%), respectively. The highest prevalence of Paramphistome infections was observed in Lupembelwasenga village (80.0%, 95% CI: 51.9-95.7%).

Figure 3. Administrative map of Tanzania showing the distribution of trematode infections in the study villages in Misungwi, Bariadi and Iringa rural districts, Tanzania.

Figure 4.The prevalence of trematode infections (F.gigantica, Paramphistomidae, and S. bovis) by region and animal species.

3.5. Fecal egg counts

The mean (±SE) of Fasciola, Paramphistomidae and Schistosoma egg counts per gram of feces in cattle were 2.4 (±)0.06, 3.0 (±)0.06, and 0.8 (±)0.05, respectively, with ranges of 0.7 to 6.1, 0.7 to 6.3, and 0.7 to 1.4. In goats, the mean epg were 2.5 (0.14), 2.6 (0.09), and 0.8 (0.10), with ranges of 0.7 to 5.9, 0 to 6.2, and 0.7 to 1.4. For sheep, the averages were 2.6 (0.11), 2.9 (0.10), and 0.9 (0.09), with ranges of 0.7 to 6.6, 0.7 to 6.5, and 0.7 to 1.4, respectively.

In goats, there was a significant correlation between age and the average epg for Fasciola (P<0.011). For Paramphistomidae infections, the mean egg count showed a significant correlation with body condition (P<0.028). Paramphistome infections had a higher mean fecal egg count per gram of feces than Fasciola and Schistosoma infections.

  1. Discussion

The burden of trematode infections of F. gigantica, Paramphistomidae and S. bovis in domesticated ruminants in various ecological zones (The Lake Victoria Zone (Mwanza and Simiyu regions) and Southern Highlands (Iringa region) of Tanzania has been investigated in this study.

Cattle showed a higher prevalence of S. bovis, Paramphistomidae and F. gigantica infections compared to small ruminants, which is consistent with findings from the study conducted in Côte d’Ivoire by Kouadio et al.,[8]. This could be attributed to the fact that goats primarily consume leaves and heaths in elevated areas, while sheep graze on open land, and cattle graze near water bodies. As a result, cattle are at a greater risk of being exposed to water and vegetation where the infective parasite stage is present, leading to infection.

The study found that Paramphistomidae infections had the highest mean fecal egg count per gram of feces (epg) followed by F. gigantica and S. bovis. A similar pattern of infection load for these three trematodes has been documented in various regions of Tanzania and other parts of Africa [34, 41]. The lack of control programmes against trematode infections leads to livestock owners medicating their domesticated ruminants with commonly used anthelminthic drugs such as albendazole and nitroxinil without a prescription from a Veterinary Officer. This may lead to the administration of incorrect dosage which accelerates the emergence of resistant trematodes parasite populations which in turn contribute to high infection intensity infections which maintains the parasite transmission cycle.

It was found that Paramphistomidae and F. gigantica infections are the predominant trematode infections in cattle, goats, and sheep in the studied areas. This aligns well with earlier studies conducted in Tanzania and other parts of Africa [7, 34, 38]. The high prevalence of Paramphistomidae infections could be attributed to the fact that the adult parasite is highly productive and capable of surviving in the host for extended periods. Also, they have wider range of snails intermediate host compared to Fasciola and Schistosoma [42]. Paramphistomidae parasites are known for their high reproductive capacity and their ability to thrive in harsh conditions [43]. Additionally, many broad-spectrum anthelmintics, such as albendazole and triclabendazole, which are commonly used to treat nematode and trematode infections have minimal or no effect on Paramphistomidae infections [44].

In regards to the low prevalence of S. bovis infection, the sedimentation technique has been shown to have a low sensitivity depending on the protocol. This may be contributed by factors such as variation in the distribution of eggs within a single fecal specimen; daily fluctuations of fecal production and consistency in the host and daily fluctuations related to oviposition patterns of the parasite [45–47].

The time samples are left to sediment affects sensitivity because of the rapid hatching of Schistosoma eggs which may occur during the sedimentation process before the sediment is observed under the microscope [48]. In fact, after exposure to water, Schistosoma eggs can hatch within 20 minutes [49]. In this study, the total time eggs were left to sediment in water was 45 min. It may also be because not all Schistosoma eggs are excreted in the feces, many are left trapped in tissue [50]. Lastly, the host immune response against Schistosome parasites is directed towards the suppression of worm fecundity, resulting in reduced egg output rather than elimination of adult worms [51].

The higher prevalence of F. gigantica could be attributed to the higher transmission of Fasciola parasites in the study regions which in turn could be attributed to the suitable environment of both the parasite and the snail intermediate host. In addition, fecal examination appears to be more sensitive for Fasciola egg detection. A study conducted in Southern Ethiopia also demonstrated a higher prevalence of F. gigantica in cattle [52]. The predominant co-infection of trematodes in this study was observed between F. gigantica and Paramphistomidae parasites. This could be attributed to the similarities in the life cycles of these parasites, which both rely on lymnaeid snails as intermediate hosts [53, 54].

The observed distribution of trematode infections in this study is likely determined by the natural ecology of the study areas. The presence of water bodies such as rivers, ponds, and streams, not only provides habitats for snail intermediate snail hosts of the parasites [18], but also animal-water contact sites (for schistosomiasis) and aquatic plants that can be consumed by domesticated ruminants (fascioliasis). The observed snail intermediate host in this study includes snails from the genus Lymnaea, Bulinus and Biomphalaria. The similar sails intermediate hosts have been reported in previous study in the Southern highlands of Tanzania [18], which play essential roles in maintaining the transmission cycle of trematode infections.

In addition, the livestock management system plays an essential role in maintaining the transmission of schistosomiasis and other trematode infections of domestic ruminants. In traditional livestock management systems, domesticated ruminants graze near and drink water from rivers and ponds, depositing feces containing the parasite eggs and hence perpetuating the transmission cycle.

The results of previous studies conducted in the Southern highlands of Tanzania indicated that transmission of trematode infections is impacted by the management systems of domesticated ruminants. Indeed, the study revealed that in areas where traditional livestock management systems were practiced the prevalence of trematodes infections was high. The prevalence was moderate in large-scale dairy management systems and lowest in small-scale dairy management systems [10]. This finding aligns with a study conducted in Mali, which demonstrated that factors such as climate, the presence of water bodies, and the type of domesticated ruminant rearing systems significantly affected the prevalence of trematode infections [55].

Adult cattle were more infected by Paramphistomidae and S. bovis infections than younger animals, likely due to prolonged exposure to contaminated water while grazing. This infection pattern has also been observed in Nigeria [56] and in Tanzania [18]. Further, it could also be possible that as domesticated ruminant gets older, their immunity against trematode infections decreases and hence experience higher infection rates [57]. On the contrary, a study found a higher prevalence rate of fascioliasis in weaners than in adults [58] and hypothesized that older animals developed acquired immunity that results in increased resistance [11].

The study limitations include the small sample size for goats and sheep in comparison to cattle across all study villages. Additionally, the study did not document the treatment history, which could have impacted the prevalence of trematode infections. The identification of parasite species relied solely on egg morphology. Nonetheless, this study underscores the significance of trematode infections of veterinary and public health importance in Tanzania. It is also the first study to provide a comprehensive overview of the magnitude and distribution of these trematode infections among domesticated ruminants in the Lake Victoria Zone (Mwanza and Simiyu regions) of Tanzania.

  1. Conclusions

The study revealed that Paramphistomidae and F. gigantica infections were more prevalent trematode infections in cattle, goats, and sheep in the Lake Victoria zone (Mwanza and Simiyu) and the Southern highlands (Iringa region) of Tanzania. These findings demonstrate that trematode infections are prevalent across the two study areas, with the highest prevalence found in the Lake Victoria zone therefore effective and community-directed control interventions are needed to break the transmission cycle and hence reduce fecal egg contamination of the environment. The strategic use of anthelmintics effective against mixed trematode infections and improvements in livestock management practices such as keeping domesticated ruminants away from grazing areas heavily contaminated by trematode eggs are highly recommended.

Author Contributions: GSM, SK, JN, AK, MES, AS, and BJV conceptualized and designed the study; GSM, SK, and JN planned and conducted the fieldwork; GSM and JN collected and examined the fecal specimens; GSM analyzed the data and drafted the manuscript; and GSM, JN, AK, SK, MES, AS and BJV revised and improved the final version of the manuscript. The authors have read and approved the final version of the manuscript.

Funding: The research work reported in this publication was funded by the European Union's Horizon 2020 as part of the PREPARE4VBD research and innovation program under grant agreement No. 101000365.

Institutional Review Board Statement: This research was a part of a project called "PREPARE4VBDs" aimed at identifying, predicting, and preparing for emerging vector-borne diseases. The study adhered to the ethical approval guidelines set forth by the Medical Research Coordination Committee (MRCC) of the National Institute for Medical Research (NIMR), which serves as the national ethics review board in Tanzania (ethics clearance certificate number NIMR/HQ/R.8a/Vol. IX/3860).

Informed Consent Statement: Informed consent was obtained from the owners of all animals involved in the study.

Data Availability Statement: The data presented in this study are available on request from the corresponding author.

Acknowledgments: The authors express their gratitude to the Regional Commissioners of the Mwanza, Simiyu, and Iringa regions, the District Executive Directors of the Bariadi, Misungwi, and Iringa Rural districts, the Regional and Districts Veterinary Officers, and the Village and Subvillage leaders for their cooperation during fieldwork. Salim Bwata, Revocatus Silayo, Aruni Haruya, Martin Anditi, and Kalebu Kihongosi are acknowledged for their technical assistance during field and laboratory work.

Conflicts of Interest: No competing interests were declared.

References

  1. Mage, C.; Bourgne, H.; Toullieu, J.M.; Rondelaud, D.; Dreyfuss, G. Fasciola hepatica and Paramphistomum daubneyi: changes in prevalences of natural infections in cattle and in Lymnaeatruncatula from central France over the past 12 years. Veterinary research 2002,33(5), 439-447.
  2. Swai E.S.; Ulicky, E. An evaluation of the economic losses resulting from the condemnation of cattle livers and loss of carcass weight due to fasciolosis: a case study from Hai town abattoir, Kilimanjaro region, Tanzania. Domesticated ruminants Res Rural Dev 2009, 21(11), 186.
  3. Mas‐Coma, S.; Valero, M.A.; Bargues, M.D. Fasciola, lymnaeids and human fascioliasis, with a global overview on disease transmission, epidemiology, evolutionary genetics, molecular epidemiology and control. Advances in parasitology 2009,69, 41-146.
  4. Dreyfuss, G.; Alarion, N.; Vignoles, P.; Rondelaud. D. A retrospective study on the metacercarial production of Fasciola hepatica from experimentally infected Galba truncatula in central France. Parasitology Research,2006,98 (2), 162-166.
  5. Chen, J.; Chen, M.; Ai L.; Xu, X.N.; Jiao, J.; Zhu, T.; Su, H.; Zang, W.;  Luo, J.;  Guo, Y.;  Lv, S.;  Zhou, X.  “An Outbreak of Human Fascioliasis Gigantica in Southwest China.” PLoS ONE, 2023, 8(8).
  6. Mas-Coma, S.; Bargues, M.D.; Valero, M.A. Human fascioliasis infection sources, their diversity, incidence factors, analytical methods and prevention measures Parasitology 2018,145:1665–
  7. Lai, Y.S.; Biedermann, P.; Ekpo, U.F.; Garba, A.; Mathieu, E, Midzi, N.; Pauline, M.; N'Goran,E.K.;  Raso, G.; Assaré, R.K.; Sacko, M.;  Schur, N.; Talla, I.;   Tchuenté, L.T.;  Touré, S.;  Winkler, M.S.;  Utzinger, J,;    Vounatsou, P. Spatial distribution of schistosomiasis and treatment needs in sub-Saharan Africa: a systematic review and geostatistical analysis. Lancet Infect Dis 2015,15:927–40.
  8. Kouadio, J.N.; Evack, J.G.; Achi, L.Y.; Fritsche, D.; Ouattara, M.; Silué, K.D.; Bonfoh, B.; Hattendorf, J.; Utzinger, J.; Zinsstag, J.; Balmer, O.; N’Goran, E.K. “Prevalence and Distribution of livestockSchistosomiasis and Fascioliasis in Côte d’Ivoire: Results from a Cross-Sectional Survey.” BMC Veterinary Research 2020, 16(1): 1–13.
  9. Schillhorn van Veen, T.W.; Folaranmi, D.O.B.; Usman, S.; Ishaya, T. Incidence of liver fluke infections (Fasciola gigantic and Dicrocoeliumhospes) in ruminants in Northern Nigeria. Tropical Animal Health and Production 1980, 12, 97–104
  10. Brown, D.S. Freshwater Snails of Africa and their Medical Importance,1994 2nd Edn. Taylor and Francis Ltd, London.
  11. Mungube, E.O.; Bauni, S.M.; Tenhagen, B.A.; Wamae, L.W.; Nginyi, J.M.; Mugambi, J.M. The prevalence and economic significance of gigantica and Stilesia hepatica in slaughtered animals in the semiarid coastal Kenya. Tropical Animal Health and Production 2006,38, 475–483.
  12. Moshfe, A.; Rezaei, NS.A.; Cheraghzadeh, SR.; Arefkhah, N.; Zare, KR.; Moein, M.; Jamshidi, A. Study on Prevalence of Fascioliasis in Ruminants in Dasht Room County in Spring and Summer of 2013.Animal and Veterinary Sciences. Vol. 4, No. 2, 2016, pp. 15-18. doi:10.11648/j.avs.20160402.11
  13. Keyyu, J.D.; Monrad, J.; Kyvsgaard, N.C.; Kassuku, A.A. Epidemiology of gigantica and amphistomes in cattle on traditional, small-scale dairy and large-scale dairy farms in the southern highlands of Tanzania. Trop Anim Health Pro 2005, 37:303–314.
  14. Walker, S.M.; Makundi, A.E.; Namuba, F.V.; Kassuku, A.A.; Keyyu, J.; Hoey, E.M.; Prodohl, P.; Stothard, J.R.; Trudgett, A. The distribution of Fasciola hepatica and gigantica within southern Tanzania–constraints associated with the intermediate host. Parasitology 2008, 135:495–503.
  15. Taylor, M. A.; Coop, R. L.; Wall, R. L. (2016). Veterinary parasitology (4th ed.). West Sussex, UK: Wiley Blackwell 390–393.
  16. Mahlau, E.A. Liver fluke survey in zebu cattle of Iringa Region, Tanzania and first finding of the small fluke Dicrocoelium hospes/Loos. Bull Epizootic Dis Afr 1970, 18:21–28.
  17. Nzalawahe,; Hannah, R.; Kassuku, A.A.;  Stothard, J.R.;  Coles, G.C.; Eisler, M.C. “Evaluating the Effectiveness of Trematocides against F. giganticaand Amphistomes Infections in Cattle, Using Fecal Egg Count Reduction Tests in Iringa Rural and Arumeru Districts, Tanzania.” Parasites and Vectors 2018, 11(1): 384.
  18. Nzalawahe, J.; Kassuku, A.A.; Russell, S.J.; Coles, G.C.; Eisler, M.C. Associations between trematode infections in cattle and freshwater snails in highland and lowland areas of Iringa. Parasitology 2015, 142(11):1430-9.
  19. Iglesias-Piñeiro, J.; González-Warleta, M.; Castro-Hermida, J. A.; Córdoba, M.; GonzálezLanza, C.; Manga-González, Y.; Mezo, M. Transmission of Calicophoron daubneyi and Fasciola hepatica in Galicia (Spain): Temporal follow-up in the intermediate and definitive hosts. Parasites and Vectors, 2016, 9, 610.
  20. Lloyd, J., Boray, J. C., & Love, S. (2007). Stomach fluke (paramphistomes) in ruminants.Primefact 452, NSW DPI stomachfluke/Prime_Fact_452_Stomach_flukeparamphistomesin_ruminants.pdf,http://www.wormboss.com.au/files/pages/worms/flukes/Accessed date: 19 April 2024.
  21. Gower, C. M.; Vince, L.; Webster, J. P. ‘Should we be treating animal schistosomiasis in Africa? The need for a One Health economic evaluation of schistosomiasis control in people and their livestock’, Transactions of The Royal Society of Tropical Medicine and Hygiene,2017, 111(6), pp. 244–247. doi: 10.1093/trstmh/trx047.
  22. Toledo, R. and Fried, B. Digenetic Trematodes, Advances in Experimental Medicine and Biology. Springer. 2014, doi: 10.1007/978-1-4939-0915-5.
  23. Kassuku, A.A.; Christensen, N.O.; Monrad, J.; Nansen, P.; Knudsen, J. Epidemiological studies of S. bovisinIringa Region, Tanzania. Acta Trop 1986, 43:153–163.
  24. Nonga, E.; Mwabonimana, M.F.; Ngowi, H.A.; Mellau, L.S.; Karimuribo, E.D. “A Retrospective Survey of Liver Fasciolosis and Stilesiosis in Domesticated ruminants Based on Abattoir Data in Arusha, Tanzania.” Tropical Animal Health and Production 2009,41(7): 1377–80.
  25. Hyera, J.M.K. Prevalence, seasonal variation and economic significance of fascioliasis in cattle as observed at Iringa abattoir between 1976–1980. Bull Anim Health Pro Afr 1984,32:356–359.
  26. Komba, E.V.G.; Mkupasi, E.M.; Mbyuzi, A.O.; Mshamu, S.; Luwumba, D.; Busagwe, Z.; Mzula, A. Sanitary practices and occurrence of zoonotic conditions in cattle at slaughter in Morogoro Municipality, Tanzania: implications for public health. Tanzania J Health Res 2012, 14(2):131-8.
  27. Mellau, L.S.B.; Nonga, H.E.; Karimuribo, E.D. A slaughterhouse survey of liver lesions in slaughtered cattle, sheep and goats at Arusha, Tanzania. Res J Vet Sci 2010,3:179–188.
  28. Msanga, J.F. Prevalence and economic importance of gigantica and Stilesia hepatica in Sukuma land, Tanzania. Tanzania Vet Bull 1985,7:9–16.
  29. Swai E.S.; Mtui, P.F.; Mbise, A.N.; Kaaya, E.; Sanka, P.; Loomu, P.M. Prevalence of gastrointestinal parasite infections in Maasai cattle in Ngorongoro District, Tanzania. Domesticated ruminants Res Rural Dev 2006, 18 (8) 107.
  30. Makundi, A.E.; Kassuku, A.A.; Maselle, R.M.; Boa, M.E. Distribution, prevalence and intensity of bovis in cattle in Iringa District, Tanzania. Vet Parasitol 1998,75:59–6.
  31. Tanzania Climate, Retrieved from The Global Historical Weather and Climate Data,https://weatherandclimate.com/tanzania (accessed 29 May 2024)
  32. Mahoo, H. Improving research strategies to assist scaling-up of pro-poor management of natural resources in semiarid areas 2005, 82pp.
  33. Tanzania Climate, Retrieved from The Global Historical Weather and Climate Data,https://weatherandclimate.com/tanzania (accessed 29 May 2024)
  34. Nzalawahe,;  Kassuku, A.A.;  Stothard, J.R.;  Coles, G.C.;Eisler, M.C. “Trematode Infections in Cattle in Arumeru District , Tanzania are Associated with Irrigation” Parasites & Vectors 2014, 7:107.
  35. Thrusfield, M. Veterinary epidemiology. 2018, John Wiley & Sons. Fourth edition.
  36. Sirois, M. Principles and practice of veterinary technology 2016, 4th St. Louis, Missouri.
  37. Soulsby, E.J.L. Helminths, arthropods and protozoa of domesticated animals 1982, 7th London: Baillere Tindall.
  38. Gouvras, A.N.; Allan, F.; Kinung'hi, S.; Rabone, M.; Emery, A.; Angelo, T. Longitudinal survey on the distribution of Biomphalaria sudanica and choanomophala in Mwanza region, on the shores of Lake Victoria, Tanzania: implications for schistosomiasis transmission and control. Parasit. Vectors,2017, 10. https://doi.org/10. 1186/s13071-017-2252-z
  39. DBL-WHO: A field guide to African freshwater snails. Danish bilharziasis laboratory. Charlottenlund, Denmark: WHO collaborating Centre for Applied Malacology; 1998.
  40. Frandsen, F. and Christensen, N. An introductory guide to the identification of cercariae from African freshwater snails with special reference to cercariae of medical and veterinary importance. Acta Tropica,1984; 41,181–202.

  1. Yeneneh, A.; Kebede, H.; Fentahun, T.; Chanie, M. Prevalence of cattle flukes infection at Andassa domesticated ruminants research center in north‒west of Ethiopia. Veterinary Research Forum 2012, 3(2), 85–
  2. Dorchies, P.H. Flukes: Old parasites but new emergence. Proceedings of the XXIV World Buiatrics Congress 2006, Vol. 16 Consultada.
  3. Hansen, J.; Perry, B. The epidemiology, diagnosis and control of helminth parasites of ruminants, a hand book. Nairobi, Kenya: International Laboratory for Research on Animal Disease (ILRAD) 1994.
  4. Rolfe, P.F.; Boray, J.C.; Nichols, P.; Collins, G.H. Epidemiology of paramphistomosis in cattle. International Journal of Parasitology 1991, 21, 813–
  5. Mas-Coma, S.; Bargues, MD.; Valero, MA. Diagnosis of human fascioliasis by stool and blood techniques: update for the present global scenario. Parasitology. 2014; 141:1918–46.
  6. Valero, MA.; Panova, M.; Pérez-Crespo, I.; Khoubbane, M.; Mas-Coma, S. Correlation between egg-shedding and uterus development in Fasciola hepatica human and animal isolates: applied implications. Vet Parasitol. 2011;183:79–86.
  7. Valero, MA.; Panova, M.; Comes, AM.; Fons, R.; Mas-Coma, S. Patterns in size and shedding of Fasciola hepatica eggs by naturally and experimentally infected murid rodents. J Parasitol. 2002; 88:308–13.
  8. Giovanoli Evack, J.; Kouadio, JN.; Achi, L.; Balmer, O.; Hattendorf, J.; Bonfoh, B.; et al. Accuracy of the sedimentation and filtration methods for the diagnosis of schistosomiasis in cattle. Parasitol Res. 2020; 119:1707–12.
  9. Jones, MK.; Bong, SH.; Green, KM.; Holmes, P.; Duke, M.; Loukas, A.; et al.Correlative and dynamic imaging of the hatching biology of Schistosoma japonicum from eggs prepared by high pressure freezing. PLoS Negl Trop Dis. 2008;2: e334
  10. Melkamu, S. Study on prevalence and associated risk factors for bovine and human schistosomiasis in Bahir Dar and its surrounding areas. J Anim Res 2016,6:967–
  11. Lawrence, J.A. Schistosomamattheei in sheep: the host-parasite relationship. Res Vet Sci 1974,17:263–
  12. Chakiso, B.; Menkir, S.; Desta, M. On farm study of bovine fasciolosis in Lemo district and its economic loss due to liver condemnation at Hossana municipal abattoir, southern Ethiopia. Int J CurrMicrobiolAppl Sci, 2014 3: 1122–
  13. Food and Agriculture Organization. The Epidemiology of helminth parasites.1993 2nd ed. p. 1–30.
  14. Keyyu, J.D.; Kassuku, A.A.; Msalilwa, L.P.; Monrad, J.; Kyvsgaard, N.C.Cross-sectional prevalence of helminth infections in cattle on traditional, small-scale and largescale dairy farms in Iringa district, Tanzania. Vet Res Commun 2006, 30(1):45-55.
  15. Tembely, S.; Galvin, T.J.; Craig, T.M.; Traore, S. Liver fluke infections of cattle in Mali. An abattoir survey on prevalence and geographic distribution. Trop Anim Health Prod 1988, 20:117–
  16. Elelu, N.; Ambali, A.; Coles, G.C.; Eisler, M.C. Cross-sectional study of gigantica and other trematode infections of cattle in Edu local government area, Kwara state, north-Central Nigeria. Parasit Vectors 2016, 9:470.
  17. Zewde, A.; Bayu, Y.; Wondimu, A. Prevalence of bovine fasciolosis and its economic loss due to liver condemnation at Wolaita Sodo municipal abattoir, Ethiopia. Vet Med Int. 2019; 2019:9572373.
  18. Yatswako, S.; Alhaji, NB. Survey of bovine fasciolosis burdens in trade cattle slaughtered at abattoirs in north-Central Nigeria: the associated predisposing factors and economic implication. Parasite Epidemiol Control. 2017; 2:30–9.